# Antibiotic resistance begets more resistance: chromosomal resistance mutations mitigate fitness costs conferred by multi-resistant clinical plasmids

**Ramith R. Nair,**[1] **Dan I. Andersson,**[1] **Omar M. Warsi**[1]

**ABSTRACT** Plasmids are the primary vectors of horizontal transfer of antibiotic resistance genes among bacteria. Previous studies have shown that the spread and maintenance of plasmids among bacterial populations depend on the genetic makeup of both the plasmid and the host bacterium. Antibiotic resistance can also be acquired through mutations in the bacterial chromosome, which not only confer resistance but also result in changes in bacterial physiology and typically a reduction in fitness. However, it is unclear whether chromosomal resistance mutations affect the interaction between plasmids and the host bacteria. To address this question, we introduced 13 clinical plasmids into a susceptible *Escherichia coli* strain and three different congenic mutants that were resistant to nitrofurantoin ($\Delta nfsAB$), ciprofloxacin (*gyrA,* S83L), and streptomycin (*rpsL,* K42N) and determined how the plasmids affected the exponential growth rates of the host in glucose minimal media. We find that though plasmids confer costs on the susceptible strains, those costs are fully mitigated in the three resistant mutants. In several cases, this results in a competitive advantage of the resistant strains over the susceptible strain when both carry the same plasmid and are grown in the absence of antibiotics. Our results suggest that bacteria carrying chromosomal mutations for antibiotic resistance could be a better reservoir for resistance plasmids, thereby driving the evolution of multi-drug resistance.

**IMPORTANCE** Plasmids have led to the rampant spread of antibiotic resistance genes globally. Plasmids often carry antibiotic resistance genes and other genes needed for its maintenance and spread, which typically confer a fitness cost on the host cell observed as a reduced growth rate. Resistance is also acquired *via* chromosomal mutations, and similar to plasmids they also reduce bacterial fitness. However, we do not know whether resistance mutations affect the bacterial ability to carry plasmids. Here, we introduced 13 multi-resistant clinical plasmids into a susceptible and three different resistant *E. coli* strains and found that most of these plasmids do confer fitness cost on susceptible cells, but these costs disappear in the resistant strains which often lead to fitness advantage for the resistant strains in the absence of antibiotic selection. Our results imply that already resistant bacteria are a more favorable reservoir for multi-resistant plasmids, promoting the ascendance of multi-resistant bacteria.

**KEYWORDS** antibiotic resistance, plasmids, drug resistance evolution, multidrug resistance, conjugation, epistasis

Plasmids are extrachromosomal genetic elements that are found in a wide variety of bacteria and have gained notoriety as vectors for the horizontal transfer of antibiotic-resistant genes (1). Apart from carrying genes that impart resistance to antibiotics, plasmids can carry other genes, including those for replication, heavy metal

Address correspondence to Ramith R. Nair, ramith_nair@hotmail.com.

The authors declare no conflict of interest.

See the funding table on p. 10.

10.1128/spectrum.04206-23   **1**

tolerance (2), and post-segregational killing systems (3) that increase the fitness of the hosts in certain environments. However, many of these genes can potentially confer a cost in other environments that can lead to the extinction of the plasmid-carrying hosts (4, 5).

The fitness cost of plasmids has been a focus of research among experimental and theoretical microbiologists for several years, resulting in a detailed understanding of the role of specific genes and functions important in plasmid maintenance and spread among bacterial populations [reviewed in references (6, 7)]. These effects on the fitness of the bacterial hosts are host-dependent. For example, the plasmid RP1 was found to have different fitness effects on different *Escherichia coli* host strains (8), while a carbapenamase-encoding plasmid was found to have a lower fitness cost in *E. coli* compared to *Klebsiella pneumoniae* (9). Furthermore, a recent study with the antibiotic resistance plasmid pOXA-48_K8 showed that the host-dependent variation in fitness effects aids plasmid maintenance in bacterial communities (10). Besides the host dependence, a specific bacterial host can also exhibit varied responses to different plasmids (11). Thus, the fitness effects of plasmids are a product of the genetic properties of the plasmid as well as that of the host, and the nature of these interactions can have a profound impact on the spread and maintenance of plasmids in bacterial populations (4, 12, 13).

Clinical antibiotics target essential functions encoded by housekeeping genes in bacteria and hence the mutations that confer resistance to these antibiotics usually result in profound changes in bacterial physiology (14, 15). For example, mutations in the gene *gyrA* not only provide resistance to quinolones such as ciprofloxacin but also alter the transcriptome that modulates the expression of stress response pathways in *Salmonella* Typhimurium (16). In another case, mutations in the gene *rpsL* (which provides resistance to streptomycin) were shown to improve growth in carbon-poor conditions (17) and survival in macrophages (18). Several other studies [reviewed in references (19, 20)] have demonstrated the physiological effects of antibiotic resistance mutations in the absence of the drug. However, whether these changes in bacterial physiologies due to resistance mutations alter their interactions with plasmids remains unclear.

To address this question, we introduced 13 clinical plasmids (Table 1) into susceptible and three antibiotic-resistant backgrounds of *E. coli* MG1655 (Table S1) and determined the effect of the plasmid on the maximum exponential growth rates of the host strains in minimal media with glucose. The plasmids used for this study are clinical plasmids isolated from extended-spectrum-β-lactamase (ESBL)-producing *E. coli* and *K. pneumoniae* (21, 22). The antibiotic-resistant host strains carry clinically relevant mutations conferring resistance to nitrofurantoin, ciprofloxacin, and streptomycin (Table 2). We find that though several of the plasmids confer costs on the susceptible *E. coli*, these costs disappear in each of the resistant backgrounds we tested. Furthermore, in certain cases, the differential effect of plasmid costs in susceptible and resistant *E. coli* resulted in a fitness advantage for the resistant strain in antibiotic-free environments.

## MATERIALS AND METHODS

### Strains and media

*Escherichia coli* K12 MG1655 derivative strains (DA28100 and DA28102, both derived from parent strain DA5438), previously reported to carry chromosomal copies of fluorescent protein genes *bfp* (blue fluorescent protein) and *yfp* (yellow fluorescent protein) (23), were used as susceptible hosts for all the plasmids. Fluorescently labeled nitrofurantoin-resistant (NIT-R) strains (24) were used as NIT-R hosts for all the plasmids. Fluorescently labeled susceptible and NIT-R strains were used in competition experiments. Ciprofloxacin-resistant (CIP-R) and streptomycin-resistant (STR-R) hosts containing chromosomal point mutations in genes *gyrA* (S83L) and *rpsL* (K42N), respectively, were

**TABLE 1**  List of plasmids used in this study[c]

| Plasmid ID | Plasmid name | Size (Kb) | Resistance genes | Genbank ID | PTU[a] |
|---|---|---|---|---|---|
| P1 | pDA51104_214 | 214 | aadA5, aac(6')-Ib, aac (3)-IId, aph (6)-Id, aph(3")-Ib, bla_{TEM-1B}, mph(A), sul1, sul2, tet(A), dfrA17, cmlA1 | CP076059 | PTU-FE |
|  | pDA51104_70 | 70 | bla_{CTX-M-14} | CP076058 | PTU-FE |
| P2 | pDA33137-178 | 178 | aadA5, aph(6')-Id, aph(3")-Ib, aac (3)-IId, aac(6')-Ib, bla_{TEM-1B}, bla_{CTX-M-14}, mph(A), sul1, sul2, tet(A), dfrA17, ermB, cmlAl | CP029580 | PTU-FE |
| P3 | p4_0.1 | 138 | aadA5, aac(3')-IIa, aac(6')-Ib-cr, bla_{CTX-M-15}, bla_{OXA-1}, mph(A), sul1, tet(A), dfrA17, catB3 | CP023850 | PTU-FE |
| P4 | p4_1_1.1 | 181 | aadA2, aph(3')-Ia, aac(6')-Ib-cr, bla_{TEM-1B}, bla_{OXA-1}, bla_{CTX-M-15}, mph(A), sul1, tet(A), dfrA12, catB3 | CP023845 | PTU-FE |
| P5 | pDA51104_214 | 214 | aadA5, aac(6')-Ib, aac (3)-IId, aph (6)-Id, aph(3")-Ib, bla_{TEM-1B}, mph(A), sul1, sul2, tet(A), dfrA17, cmlA1 | CP076059 | PTU-FE |
| P6 | pDA51122_89 | 89 | bla_{CTX-M-14} | CP076056 | PTU-I1 |
| P7 | pDA51128_93 | 93 | bla_{CTX-M-1} | CP076054 | PTU-I1 |
| P8 | pDA51124_123 | 123 | aph (6)-Id, aph(3")-Ib, bla_{CTX-M-65}, fosA3 | CP076055 | PTU-I1 |
| P9 | pDA33135-70 | 70 | bla_{CTX-M-14} | CP029578 | PTU-FE |
| P10 | pDA33135-139 | 139 | aadA5, aph (6)-Id, aph(3")-Ib, aac (3)-IId, bla_{TEM-1B}, mph(A), sul1, sul2, tet(A), dfrA17 | CP029577 | PTU-FE |
| P11 | p7_2.1 | 113 | aadA5, mph(A), sul1, dfrA17 | CP023821 | PTU-FE |
| P14 | pDA33138_215 | 215 | aph(3")-Ib, aph (6)-Ide, aac(6')-IB3, aac (3)-IId, aadA5, sul1, sul2, dfrA17, tet(A), bla_{TEM-1B}, bla_{TEM-f}, cmlA1, mph?(a) | NA[c] | NA[c] |
| P15 | pDA61218_116 | 116 | aadA1, sul1, bla_{SHV-1/SHV-48/SHV-102g} | CP061207 | PTU-FE |

[a]As predicted by COPLA.
[b]First column denotes the ID used for the plasmid in this study, "Plasmid Name" is the name given to the plasmid as per information on GenBank, the rest of the columns denote the size, resistance genes, GenBank Accession number, and Plasmid Taxonomic Unit (as determined by COPLA) of the plasmids used here.
[c]Plasmid sequence unavailable.

constructed as previously described (25). A list of all the strains used in this study is shown in Table S1.

## Conjugations

Plasmids were conjugated into each of the susceptible and resistant host strains through a Δ*dapA* donor strain [cannot grow in the absence of 1,6 diaminopimelic acid (DAP)] to enable counterselection of plasmid-bearing recipient strain. The Δ*dapA* strains harboring the appropriate plasmids were constructed in-house by transducing a chloramphenicol resistance (cat) cassette into the *dapA* gene. The phenotypes for the deletions were independently verified by us. Plasmid-bearing Δ*dapA* and the host strains were streaked onto lysogeny broth (LB) agar (5 g yeast extract, 10 g tryptone, 10 g NaCl, and 15 g Agar per liter) and incubated overnight at 37°C (20 µg/mL DAP were also added to the media). Individual colonies from host and recipient strains were then inoculated into LB broth with and without antibiotics (as appropriate) and grown overnight at 32°C. Fully grown cultures were then diluted 1:100 in fresh LB media (with DAP) and grown at 37°C under shaking for 2 hours. Next, the recipient and donor strains were mixed at a ratio of 1:10, and 50 µL of the mix was spotted onto LB plates with DAP and incubated overnight at 37°C to allow conjugation. The spots were then harvested using a sterile loop and then resuspended in 1 mL phosphate-buffered saline and plated onto selective plates [LB agar without DAP and containing appropriate plasmid-specific antibiotics (Table 1)]. Individual colonies were then restreaked onto fresh selective plates (twice) to obtain pure colonies with the plasmid. Transconjugants for plasmid P15 could not be obtained in the STR-R background and hence are not included in the subsequent plot and analyses.

**TABLE 2** List of antibiotic-resistant host strains[a]

| Strain | Mutation | Resistance | Resistance mechanism |
|---|---|---|---|
| DA65117 | Δ*nfsAB* | Nitrofurantoin | Deletion of nitroreductase encoding genes, preventing the activation of the antibiotic |
| DA49828 | *rpsL* K42N | Streptomycin | Target modification of streptomycin-binding site (ribosomal protein S12) |
| DA49842 | *gyrA* S83L | Ciprofloxacin | Target modification of ciprofloxacin-binding site (DNA gyrase) |

[a]Strain name, chromosomal mutation (column 2) that gives resistance to the antibiotic (column 3), and the resistance mechanism (column 4).

## Growth measurements

Antibiotic susceptible and recipient strains with and without plasmids were streaked onto LB plates and incubated overnight at 32°C. Six independent cultures were then started in minimal media (M9 salts) with 0.2% glucose and grown overnight under shaking at 32°C. Overnight grown cultures were then diluted 1:100 in fresh minimal medium. Three hundred microliters of this suspension was inoculated into the wells of the honeycomb plates and the change in optical density (OD) at 600 nm was measured every 4 min using a Bioscreen C reader (Oy Growth Curves Ab Ltd). The plates were incubated in the Bioscreen C analyser at 32°C with shaking for 24 h. The exponential phase growth rate was obtained using the package growthcurver (26) in R. Relative growth rate was obtained for each strain by dividing the respective values for each plasmid-bearing strain with that obtained for the respective host strain. Relative growth rates were determined for each replicate individually and then averaged to generate Fig. 2. In addition, growth rates of each host background strain relative to the wild-type MG1655 were also independently determined (Fig. S1). The topmost well in each column of the plate was inoculated with control strains and the exponential growth rate from that well was used to calculate the relative growth rate for all the wells below it.

## Competitions

Competition experiments were performed to determine the relative fitness of plasmid-bearing nitrofurantoin susceptible and resistant *E. coli* using fluorescently (*bfp* and *yfp*) labeled strains. Six colonies of each bacterial strain were grown in minimal media with 0.2% glucose. Five plasmids that gave the highest relative growth rate (on average) in the NIT-R host background were chosen for these competitions. For each plasmid, the susceptible strain tagged with one of the two fluorescent markers was mixed at 1:1 with the NIT-R strain carrying the other marker in the same media, thus generating 12 biologically independent competition mixes. Two microliters of the *E. coli* culture mix was added to a well in a 96-well plate containing 198 μL of sterile minimal media to start the competition. The competitions were performed under shaking at 32°C for 24 h. The ratio of resistance to sensitive cells was measured by counting at least $10^5$ cells using a fluorescence-activated cell sorter (BD FACS Aria) at the start and the end of the competition. The relative fitness of the resistant strain was measured using the formula of Ross-Gillespie et al. (27):

$$v = \frac{x2(1-x1)}{x1(1-x2)}$$

where $x1$ and $x2$ are the initial and final frequencies of the NIT-R cells, respectively.

The percentage of resistant cells as well as relative fitness was averaged across all 12 independent biological replicates.

## Minimum inhibitory concentration testing

The minimum inhibitory concentration (MIC) for nitrofurantoin, ciprofloxacin, and streptomycin for all the corresponding resistant strains was determined using Etests, as per the instructions from the manufacturer (bioMérieux, Marcy l'Étoile, France). Briefly, overnight-grown cultures grown in minimal glucose media were diluted 1:10 in fresh sterile media, and then 100 μL was spread on minimal glucose agar plates using sterile

loops. The Etests were then placed on the plates after which these were incubated at 32°C for 24 h. Two biological replicates were done in each case.

## Plasmid sequence analysis

Plasmid sequences were downloaded from the GenBank website as fasta files. The files were then used to sketch a combined archive of all the genomes and calculate the pairwise Mash distance of each genome from all the genomes in the combined archive using Mash 2.0 (28). The calculated Mash distances were then used to generate a distance matrix (submitted as one of the data files), which was used to generate the neighborhood-joining tree using ggdendro (29) and ggplot2 (30). Plasmid Taxonomic Units (PTU) for all the plasmids were determined using the web interface of COPLA (31) at *https://castillo.dicom.unican.es/copla/*.

## Statistics

Relative growth rates for each plasmid were calculated as the ratio of the exponential growth rate of the plasmid-bearing strain over that of the plasmid-free host strain for each replicate. The effect of host genetic background on the fitness costs conferred by plasmids was tested with one-way ANOVA by designating host genetic background as the independent variable and mean relative growth rate for each population as the dependent variable (Fig. 1). The overall average for each host background was tested against 1 using Welch's one-sided *t*-test with and corrected for multiple testing with false discovery rate.

Within each genetic background, the relative growth rate estimates for each plasmid were tested against 1 using Welch's one-sided *t*-tests to determine the statistical significance of the fitness effect of each plasmid on its host (Fig. 2; Table S2). The *P* values were adjusted using false discovery rate correction to identify plasmids causing significant fitness changes in the respective hosts.

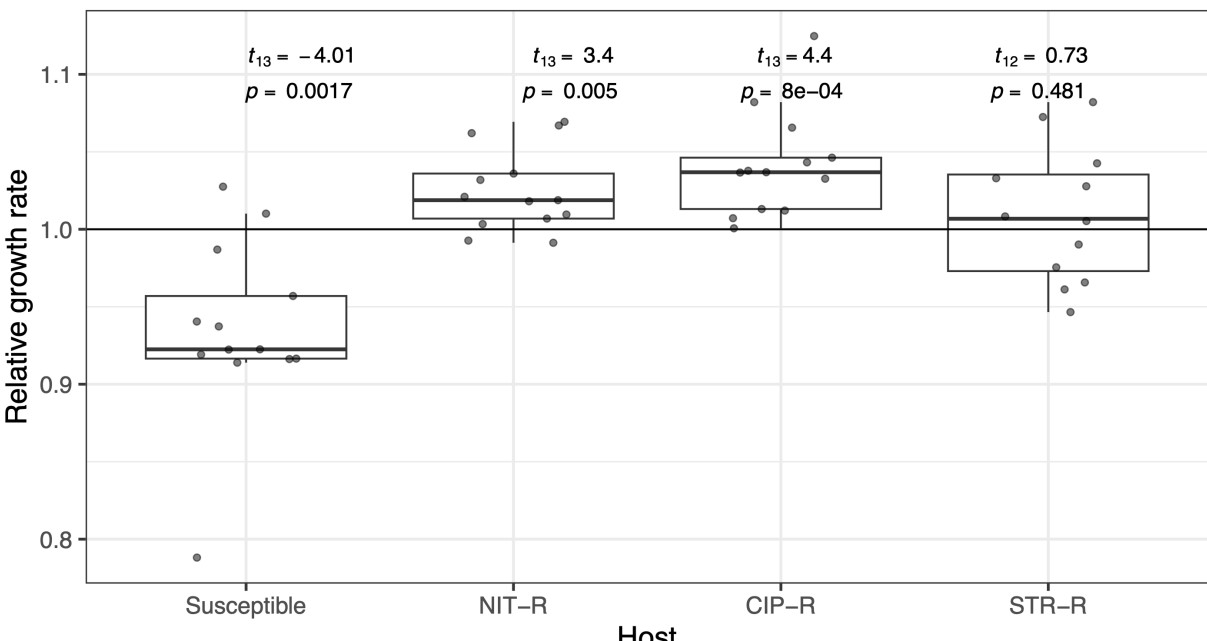

**FIG 1** Plasmids reduce exponential growth rates of a susceptible strain but not for resistant strains. Box and whiskers plot depicting the distribution of relative exponential growth rates of plasmid-bearing strains across the four host genetic backgrounds. The darker lines depict the median relative exponential growth rate for each genetic background. Numerical values above each plot represent those after Welch's one-sample two-sided *t*-test of each distribution against 1. A value of 1 (horizontal line) represents no change in growth rate among plasmid-bearing strains when compared to the plasmid-free one, while a value less than 1 indicates fitness cost conferred by the plasmid.

Relative fitness values in all cases were log$_{10}$-transformed to achieve normal distribution. The average log-transformed relative fitness values thus obtained were tested against zero using Welch's one-sided $t$-tests (Fig. 3). The $P$ values were corrected for multiple tests using false discovery rate correction to identify the plasmid-bearing NIT-R strains that are significantly competitive against the NIT-S strains bearing the same plasmid.

Statistical analyses were performed in R (32) using RStudio 2023.09.1+494. All graphs were made using the package ggplot2 (30).

## RESULTS

### Clinical multi-resistance plasmids confer no cost on antibiotic-resistant mutants in nutrient-poor media

Multi-resistance plasmids isolated from *Escherichia coli* and *Klebsiella pneumoniae* clinical strains producing ESBL were conjugated into susceptible *E. coli* MG1655 background and strains containing chromosomal mutations for resistance to nitrofurantoin (NIT, Δ$nfsAB$), ciprofloxacin (CIP, $gyrA$ S83L), and streptomycin (STR, $rpsL$ K42N) (Table 2). Exponential growth rates (relative to the wildtype *E. coli* MG1655) were found to vary for different host backgrounds (Fig. S1, one-sided ANOVA: $F_{3,32} = 39.66$, $P = 6.85 \times 10^{-11}$); however, significant costs (~25%) were only observed for the STR-resistant strain (Welch's one-sample two-sided $t$-test: $t_6 = -12.7$, $P = 0.0002$, adjusted with FDR correction) in concurrence with previously published results (17). Relative growth rates were similar for susceptible, NIT-, and CIP-resistant strains (Tukey's HSD test: NIT-R vs susceptible, $P = 0.96$; CIP-R vs susceptible, $P = 0.97$; STR-R vs susceptible, $P = 3.19 \times 10^{-10}$).

The fitness effect of plasmids was measured as a ratio of the exponential growth rate of the plasmid-carrying strain over the corresponding isogenic strains without the

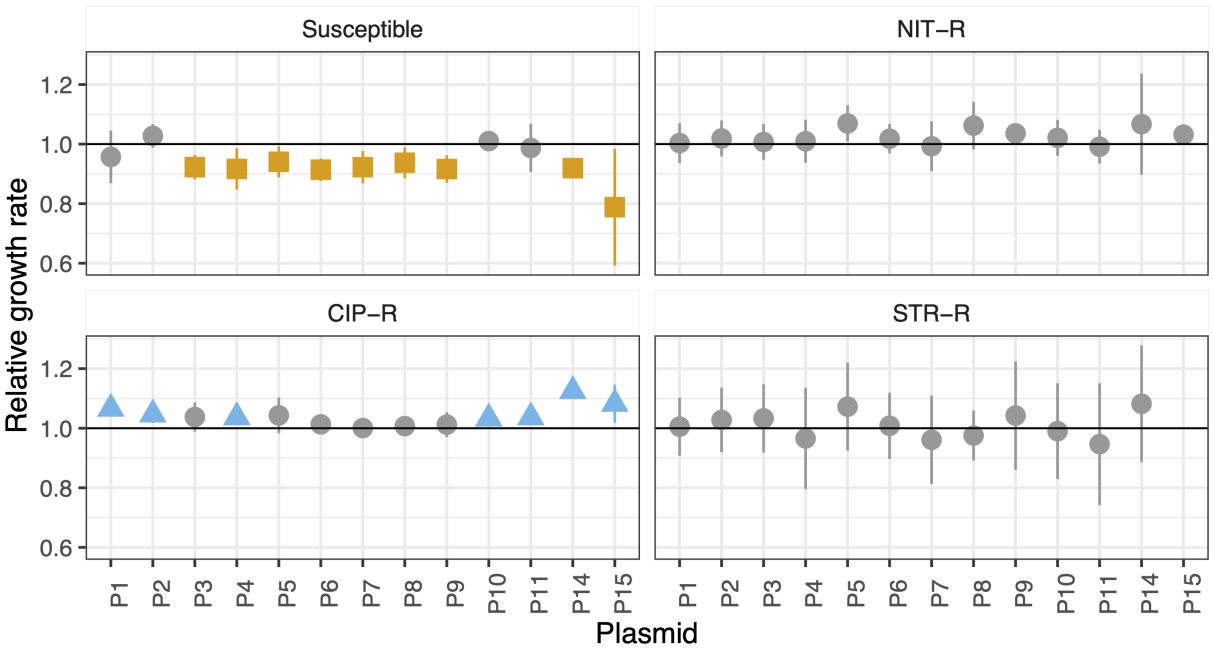

**FIG 2** The effect of plasmid carriage on bacterial relative growth rates depends on the plasmid and the host background. Relative exponential growth rates for each plasmid across the four genetic backgrounds, (A) susceptible, (B) NIT-R, (C) CIP-R, (D) and STR-R. A value of 1 (horizontal line) represents no change in growth rate for plasmid-bearing strain when compared to the plasmid-free one, while a value less than 1 indicates that a fitness cost is conferred by the plasmid. Orange squares depict relative exponential growth rates significantly lower than 1, blue triangles depict values higher than 1, and gray circles depict values equal to 1 (following Welch's one sample two-sided $t$-tests against 1). Points represent means and error bars represent 95% confidence intervals [$t$-distribution, $n = 4$–11 (Table S2)].

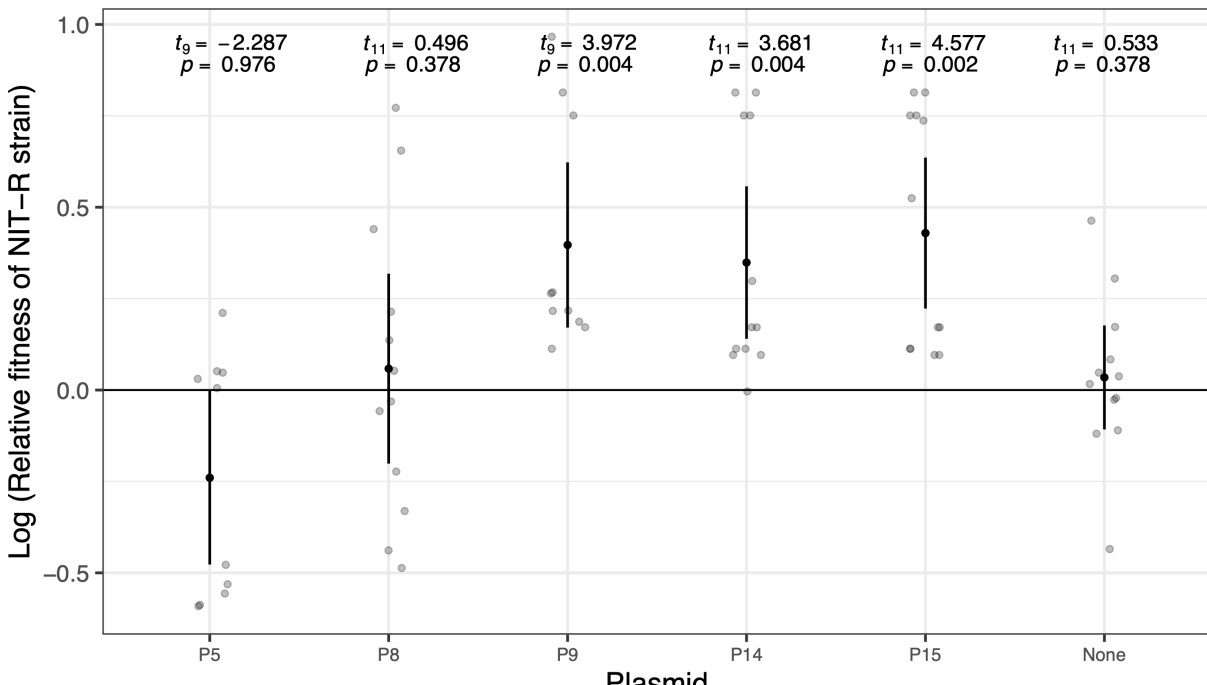

**FIG 3** Differences in relative exponential growth rates translate to relative fitness advantages for plasmid-carrying NIT-R strains. Depiction of log-transformed relative fitness of plasmid-bearing NIT-R cells after competition with NIT-S cells carrying the same plasmid for five different plasmids. Log-transformed relative fitness value of zero represents no fitness advantage to either NIT-R or NIT-S strains, while positive values indicate an advantage to NIT-R strains. Lighter points depict individual replicates, darker points represent means, and error bars represent 95% confidence intervals (*t*-distribution, *n* = 12).

plasmid. Host background had a significant effect on the observed plasmid exponential growth rates (one-sided ANOVA: $F_{3,47} = 15.47$, $P = 2.69 \times 10^{-7}$). Overall, plasmids conferred a cost (average relative growth rate = 0.936) on the exponential growth rate of susceptible strains, while no such costs were detected in the antibiotic resistance backgrounds (Fig. 1, average relative exponential growth rates in- NIT-R = 1.017, CIP-R = 1.033, STR-R = 0.996).

The negative effect on exponential growth rates of susceptible hosts was not uniform for all the plasmids tested (one-sided ANOVA: $F_{14,85} = 2.792$, $P = 0.002$). Significant costs were observed for 9 out of the 13 plasmids and the highest cost of 21% was observed for plasmid P15 (Fig. 2; Table S1). In striking contrast, none of the plasmids had any negative effects on the growth rates of antibiotic-resistant strains (Fig. 2B; Table S2). Instead, seven plasmids had statistically significant positive effects on the growth rates of CIP-R strains, which includes the four plasmids that did not confer a cost on susceptible hosts (Fig. 2A). Plasmids did not have a uniform effect on the fitness of the four *E. coli* strains (Fig. 2; Fig. S2). In addition, apart from one case with NIT, plasmids did not change the MIC of the resistant strains to the respective antibiotics (Table S3).

## Plasmid-mediated effects on growth rates confer an advantage for resistant strains in competitions

Differential effects on growth rates of susceptible and resistant strains due to plasmid carriage might result in an advantage for resistant strains in head-to-head competitions. To examine this idea, we selected five plasmids with the highest exponential growth rates (on average) in the NIT-R strain (Fig. 2) and competed them against susceptible strains carrying the same plasmid. Plasmid-carrying NIT-R strains were found to be fitter than the corresponding susceptible strains carrying the same plasmid in three out of the five cases (Fig. 3, plasmids P9, P14, and P15). NIT-R strains carrying plasmid P8 did not affect the relative fitness, while the value for those carrying plasmid P5 was negative

on average. However, it is worth noting that among the five plasmids selected for this experiment P5 and P8 had the highest average relative growth rates of 0.94 and 0.937, respectively, in susceptible strains (Fig. 2).

## Plasmids that confer cost on susceptible hosts are genetically dissimilar

To examine whether the effects of the plasmids in susceptible and resistant backgrounds were correlated to close genetic relatedness among the plasmids, we compared the sequence information of 12 plasmids for which genetic information is available (Table 1).

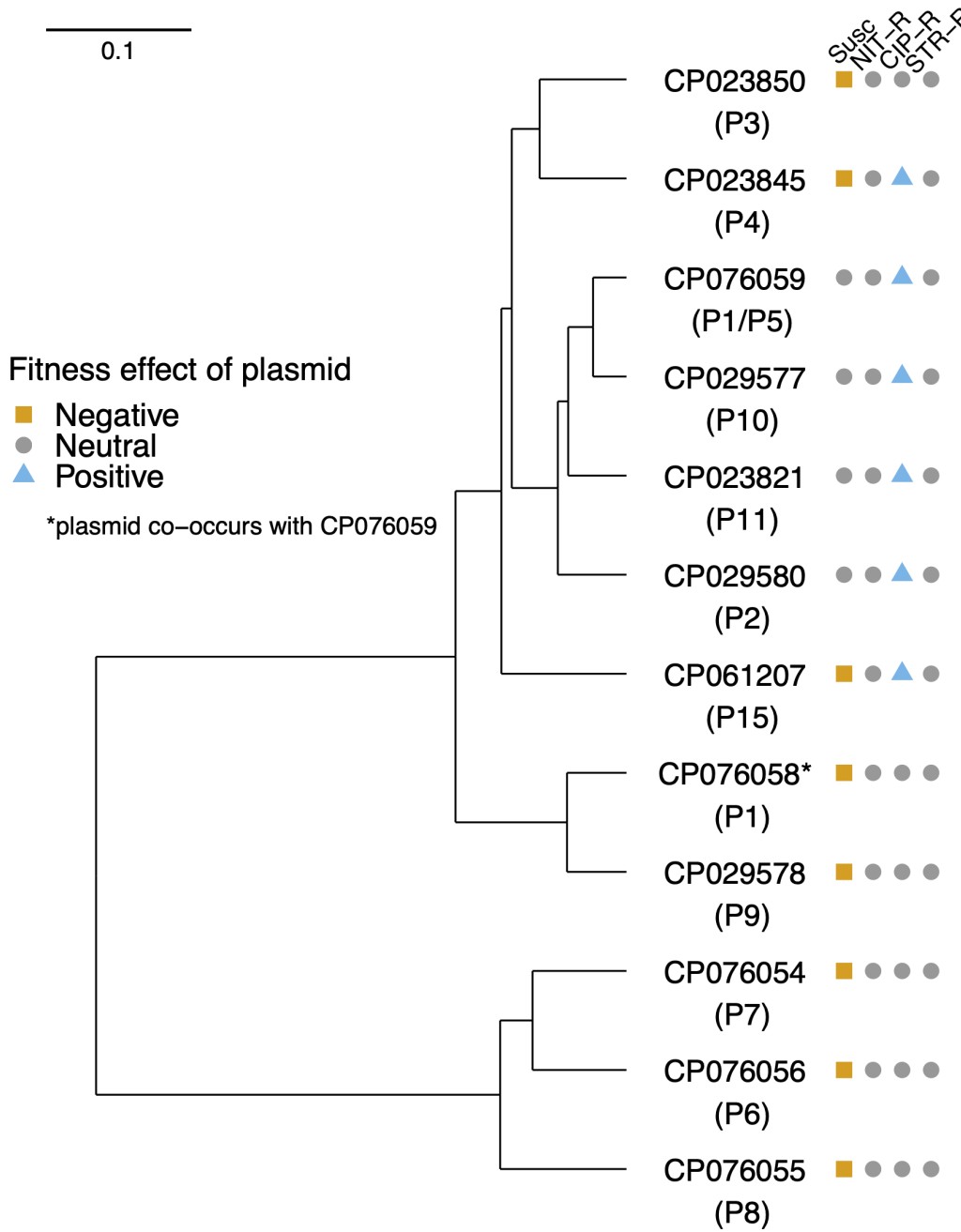

**FIG 4** Neighborhood-joining tree for the 12 plasmids. Neighborhood-joining tree based on Mash distances between plasmid sequences. Leaves on the tree indicate the GenBank accession number for each sequence with the corresponding plasmid ID in parentheses below. Points, squares, and triangles represent the same as that in Fig. 2. The distance matrix used to generate the tree is available as part of the raw data files.

We calculated genetic relatedness between the plasmids by calculating Mash distances between the plasmid sequence pairs (28) and using the same to generate a neighbor-hood-joining tree (Fig. 4). The four plasmids that did not confer costs in the susceptible hosts are grouped into a separate branch on the tree, whereas in contrast, the plasmids that did confer costs were spread over the tree.

Sequence information is available for six of the seven plasmids that conferred a positive effect on the fitness of CIP-R hosts. As noted above, four of them are the same that confer no cost on the susceptible hosts (Fig. 2) and are grouped in a separate branch while the remaining two plasmid sequences that confer fitness advantage are positioned on two distinct branches. Five out of the six sequences that confer no cost on CIP-R hosts are separated into a distinct branch.

Finally, we also looked at the correlation between relative growth rates and the size of plasmids. Plasmid-mediated fitness effects were not correlated to its size in susceptible or in NIT-R and STR-R backgrounds, while a weak positive correlation was seen in the CIP-R background [Fig. S3, Pearson's $r$ for susceptible = 0.339 ($P$ = 0.3), NIT-R = −0.211 ($P$ = 0.5), CIP-R = 0.575 ($P$ = 0.06), STR-R = −0.003 ($P$ = 0.99)]. Taken together, our results demonstrate that the distinct fitness effects of plasmids on CIP-R hosts could be due to genetic relatedness to a certain extent, while the effect on the susceptible host seems to hold for a variety of plasmid sequences and is not linked to genetic relatedness among the plasmids tested here.

## DISCUSSION

Consistent with previously published results (4, 8–11), we find that the fitness effects of plasmid carriage are dependent on the plasmid and the genetic composition of the host strain (Fig. 2). Furthermore, and most importantly, we show that in a minimal glucose medium, the 13 examined plasmids in most cases confer a fitness cost on wild-type susceptible *E. coli* host but that these costs are often abrogated when the host strains have resistance mutations on the chromosome (Fig. 1 and 2). Rajer and Sandegren (21) tested the fitness cost for 11 of the plasmids used for this study in rich media (Mueller-Hinton broth) at 37°C and found that none of them conferred any significant costs on the susceptible *E. coli* host (21). Interestingly, they found that two of them (plasmids P5 and P9 in this study) improved the fitness of the susceptible host strain in the rich media. Taken together, these results show that the fitness effects of plasmids are highly dependent on the specific plasmid, the genetic background of the host bacteria, and the growth conditions, in a presently unpredictable way. However, it should be noted that the limitations of our study are that it only includes strains of a single bacterial species carrying a limited number of plasmids grown in a single nutrient-limited medium under laboratory conditions.

The resistant bacterial strains used in this study carried chromosomal mutations that rendered them resistant to nitrofurantoin, ciprofloxacin, and streptomycin (Table 2). These mutations affect very different aspects of cellular physiology and it is difficult to explain how they can have the common effect of mitigating the fitness costs of plasmids. Nitrofurantoin resistance is gained through the deletion of nitroreductase encoding genes (*nfsA* and *nfsB*) in *E. coli* (24, 33). Apart from the fact that these oxygen-insensitive reductases are involved in the activation of pro-drugs such as nitrofurantoin, little is known about the role of these enzymes in *E. coli* (34). The ciprofloxacin-resistant strain carries a clinically relevant mutation (S83L) in the quinolone resistance determining region (QRDR) region of the DNA gyrase encoding gene (*gyrA*) (35). Mutations in *gyrA* conferring quinolone resistance have been shown to module transcriptomic profile in *Salmonella* that rendered the bacterial population resistant to multiple drugs and altered the expression profile of stress-related genes (16). In another study, a mutation in the *gyrA* gene was shown to improve *E. coli* survival in macrophages (18). The streptomycin resistance is conferred by a clinically relevant point mutation (K42N) on a gene encoding ribosomal protein S12, *rpsL* (36). Mutations in *rpsL* can modulate the fitness of *E. coli* when grown in media with different carbon sources. The specific mutation tested for this

study (*rpsL* K42N) was found to be disadvantageous (relative to the susceptible strain) in growth media with glucose and glycerol as sole carbon sources but held a fitness advantage with pyruvate and succinate (17). Furthermore, *rpsL* mutants were found to have enhanced late-growth phase protein synthesis capabilities (37), improved survival inside macrophages (18, 38), and made them more susceptible to oxidative stress (39). Based on the above findings, one could speculate that changes in bacterial physiology emanating from resistance mutations could be affecting interaction with plasmids. For example, changes in the bacterial gene expression due to the resistance mutations could reduce levels of plasmid replication functions (and plasmid copy number) and/or expression of costly plasmid-encoded genes (40, 41), similar to the gene silencing previously observed for some plasmids. Further work is required to test these hypotheses.

In conclusion, the results presented here suggest that antibiotic-resistant bacteria might be a better host reservoir for plasmids as compared to susceptible ones because they can mitigate the fitness costs of plasmids by an as yet undetermined mechanism. As a result, following the spread of plasmids in a population, resistant strains might outcompete the susceptible strains even in the absence of antibiotics, which could result in a faster appearance, spread, and stable maintenance of multi-resistant bacteria with a combination of chromosomal resistance mutations and resistance plasmids.

## ACKNOWLEDGMENTS

We thank Linus Sandegren, Marie Wrande, Hervé Nicoloff, and Jennifer Jagdmann for their helpful discussions as well as for sharing the clinical plasmids used in this study.

This work was supported by grant KAW 2018.0168 from the Wallenberg Foundation (DIA), grant 2021-02091 from the Swedish Research Council, Medicine and Health (DIA), and grant 2021-04831 from the Swedish Research Council (OMW).

R.R.N. performed all the experiments, collected and analyzed the data, and wrote the initial draft. R.R.N., D.I.A., and O.M.W. planned the project and co-wrote the final manuscript.

## AUTHOR AFFILIATION

[1]Department of Medical Biochemistry and Microbiology, Uppsala University, Uppsala, Sweden

## AUTHOR ORCIDs

Ramith R. Nair  http://orcid.org/0000-0003-2327-3813
Dan I. Andersson  http://orcid.org/0000-0001-6640-2174
Omar M. Warsi  http://orcid.org/0000-0003-3175-1184

## FUNDING

| Funder | Grant(s) | Author(s) |
| --- | --- | --- |
| Knut och Alice Wallenbergs Stiftelse (Knut and Alice Wallenberg Foundation) | KAW 2018.0168 | Dan I. Andersson |
| Vetenskapsrådet (VR) | 2021-02091 | Dan I. Andersson |
| Vetenskapsrådet (VR) | 2021-04831 | Omar Warsi |

## AUTHOR CONTRIBUTIONS

Dan I. Andersson, Conceptualization, Funding acquisition, Supervision, Writing – review and editing | Omar M. Warsi, Funding acquisition, Supervision, Writing – review and editing.

## DATA AVAILABILITY

Raw data used in this study are available through Figshare (https://doi.org/10.6084/m9.figshare.25400194).

## ADDITIONAL FILES

The following material is available online.

### Supplemental Material

**Supplemental material (Spectrum04206-23-s0001.pdf).** Tables S1 to S3 and Figures S1 to S3.

### Open Peer Review

**PEER REVIEW HISTORY (review-history.pdf).** An accounting of the reviewer comments and feedback.

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
