## [Reviewer comments · Microbiology Spectrum]

Microbiology Spectrum

Antibiotic resistance begets more resistance: Chromosomal resistance mutations mitigate fitness costs conferred by multiresistant clinical plasmids

Ramith Nair, Dan Andersson, and Omar Warsi

Corresponding Author(s): Ramith Nair, Uppsala Universitet

Review Timeline:

Submission Date:	December 18, 2023
Editorial Decision:	February 12, 2024
Revision Received:	February 23, 2024
Accepted:	March 8, 2024

Editor: Sen Pei

Reviewer(s): Disclosure of reviewer identity is with reference to reviewer comments included in decision letter(s). The following individuals involved in review of your submission have agreed to reveal their identity: Zhen Shen (Reviewer #2)

Transaction Report:

DOI: <https://doi.org/10.1128/spectrum.04206-23>

Re: Spectrum04206-23 (Antibiotic resistance begets more resistance: Chromosomal resistance mutations mitigate fitness costs conferred by multiresistant clinical plasmids)

Dear Dr. Ramith R Nair:

Thank you for the privilege of reviewing your work. Below you will find my comments, instructions from the Spectrum editorial office, and the reviewer comments.

Your manuscript has been reviewed by three experts, who provided constructive comments on the work. We would like to invite a revised manuscript that addresses their questions.

Revision Guidelines

Sincerely,
Sen Pei
Editor
Microbiology Spectrum

Reviewer #1 (Comments for the Author):

The authors addressed whether chromosomal antimicrobial-resistance mutations affect the cost of plasmid-carriage by experiments, then showed that one E.coli host carrying gyrA QRDR mutation improves fitness after plasmid acquisition at least under a defined culture condition. I think that this study provides a novel insight into the mechanisms by which QRDR mutations

become common among strains carrying plasmids (or the other way around).

However, to be fair I think the authors should state limitation of this study: for example, small sample size, limited medium condition, and unclear molecular mechanisms.

I am wondering if the authors forgot adding information for the background of model strain MG1655. MG1655 is a K-12 derivative strain that lost F plasmid, thus MG1655 already coevolved with F-type plasmid. It might not grow efficiently without F-type plasmids. How are the plasmids P1-P15 different from the F plasmid? This might be associated with the pattern in Figure 2 top left. I recommend the authors add F plasmid to the Mash distance tree.

Below I listed other points that need clarification or correction.

Line 91: MG1655 "derivative" strains. "derivative" should be added.

Line 91: I think that strain table containing full strain list rather than table 2 is needed for clarity.

Table 1: family name and allele number of bla genes in subscript should not be italicised.

Table 1: I think that this table requires more information for each plasmid, especially plasmid group that each plasmid belongs.

Please try Mob-typer for MOB group (<https://github.com/phac-nml/mob-suite>) or COPLA for PTU

(<https://bmcbioinformatics.biomedcentral.com/articles/10.1186/s12859-021-04299-x>; <https://castillo.dicom.unican.es/copla/>).

Otherwise some readers may not show interests to the findings of this study.

Line 101: Please describe the exact name of delta dapA strain. BW29427 or beta3914? Please update table 2 to include all strains used.

Line 118: It is not clear why 32DegC was used instead of 35DegC (CLSI guideline) or 37DegC (optimal condition for E.coli).

Line 119: "g"lucose.

Line 133: Is "bfp" identical to "cfp"? "cfp" is more common, I think.

Line 144: The relative fitness "(Nu)" of. The relative fitness is often expressed by "W" in evolutionary biology. The readers might think why Nu.

Line 171, 179, and Figure 1 legend: this "one-sided t-test" should be "one sample t-test". Since author cannot have a hypothesis for the results (positive effect or negative effect on fitness), two-tailed test is more appropriate. Therefore, "two-tailed one-sample t-test" is more appropriate.

Line 188: I think that the readers might want a brief introduction of these plasmids regarding the plasmid groups they belong.

Table 2: Please list all strains including non-resistant strains used. Otherwise, the method part is difficult to track.

Line 227/Table 1: It would be nice for readers to see gene content variation among plasmids in Roary-type presence/absence matrix as supplementary figure or main figure.

Line 228: Please show scale bar for the mash distance tree in figure 4.

Line 228: Please show Plasmid ID in addition to Accession number in figure 4.

Line 270: clinically relevant mutation. This must be one of so-called "QRDR mutations" in clinical microbiology. Please state so, and cite one relevant reference.

Line 270: gyrA should be italicized.

Reviewer #2 (Comments for the Author):

Ramith et al. present a specific issue about the interaction between resistance plasmids and bacterial hosts. They demonstrated that chromosomal resistance mutations could mitigate fitness costs conferred by multi-resistant clinical plasmids. This study is very interesting, and I just have several concerns.

1. What is the prevalence of chromosomal resistance mutations (Δ nfsAB, gyrA S83L, and rpsL K42N) in clinical isolates, such as E. coli and K. pneumoniae? Are there disparities in the prevalence of resistance plasmids among clinical isolates with or without chromosomal resistance mutations?

2. Clinical isolates harboring resistance plasmids typically undergo numerous generations of co-evolution, whereby the fitness cost associated with these plasmids may be alleviated through the modulation of gene expression. Following the transfer of plasmids into the recipient strain as described in this study, did the authors attempt successive passages of the strains before assessing the fitness cost?

Reviewer #3 (Comments for the Author):

The authors measure the fitness effects of 15 different MDR plasmids in sensitive E. coli and E. coli which harbour chromosomal mutation providing resistance to various antibiotics to determine if host resistance impacts the costs of plasmid carriage. The authors find that MDR plasmids are more likely to impose significant costs in sensitive hosts relative to resistant hosts. This results in a competitive advantage to resistant strains carrying MDR plasmids relative to sensitive strains harbouring

the same plasmids. The authors find that plasmids that did not impose a cost in the sensitive background were genetically closely related, whereas plasmids that imposed costs were genetically diverse. In ciprofloxacin resistance background these same plasmids provided fitness benefits. There was no clear correlation with plasmid size and fitness effects of the plasmids.

General comments:

Is it not clear that the antibiotic sensitive, NIT-R and constructed ciprofloxacin and streptomycin resistant strains are isogenic, originating from the same parent strain. If they are not isogenic variants other than the resistance mutation may be influencing the fitness in the presence of the plasmids. If not isogenic it is impossible to state if the presence of the chromosomal resistance have anything to do with the observed effects of the plasmids in these backgrounds.

Mutations that ameliorate the cost of plasmids can happen very rapidly (see, DOI: 10.1099/mic.0.000862). Can you confirm this did not occur during the conjugation of the clinical plasmids into the E. coli strains? Given that the chromosomal resistant strains already harboured costs, the selective pressure to ameliorate the plasmids may have been even greater.

I would have liked to have seen the growth of each of the E. coli backgrounds relative to each other, how much of a fitness effect do the chromosomal resistance mutation impose on the hosts in the absence of the plasmids? This will help to set the results of the effect of the plasmid in context. Do the plasmids have little effect on growth of the resistant strains because they are already very unfit?

What is the correlation in fitness effects of the plasmids across the different backgrounds? Are the same plasmids more likely to impose fitness benefits across all resistant backgrounds?

Could other properties of the MDR plasmids correlate with fitness (e.g. number of resistance genes, of the presence of specific resistances?)

Specific comments:

Line 48, plasmids can or may carry other genes

Line 190-192, are the strains containing chromosomal mutations providing resistance to NIT, CIP and STR isogenic with the drug sensitive MG1655?

Line 214-215, The selection criteria does not seem to correlate with the plasmids selected: states that they selected the plasmids with the highest exponential growth rates in the NIT-R strain (P5, P8, P9, P14 and P15) but P5 P8 and P9 in figure 2 do not have a significant increase in growth rate relative to plasmid free. Why select these over say P1, P2 or P4?

Line 218, P8, if it's not significant then it did not have a fitness advantage.

Line 221-222, do you mean relative to P9, P14 and P15 and not out of all the 15 plasmids? They still imposed a significant cost in the sensitive strain.

Figure 2, why isn't data available for P15 in STR-R strain?

Figure 4, put the plasmids ids used in the other figures on the nodes of the tree to make it easier to compare between data sets. The tree needs a scale.

RESPONSE TO REVIEWERS

Antibiotic resistance begets more resistance: Chromosomal resistance mutations mitigate fitness costs conferred by multiresistant clinical plasmids

Ramith R. Nair, Dan I. Andersson and Omar M. Warsi

Reviewer #1 (Comments for the Author):

The authors addressed whether chromosomal antimicrobial-resistance mutations affect the cost of plasmid-carriage by experiments, then showed that one E.coli host carrying gyrA QRDR mutation improves fitness after plasmid acquisition at least under a defined culture condition. I think that this study provides a novel insight into the mechanisms by which QRDR mutations become common among strains carrying plasmids (or the other way around).

We thank the reviewer for reading our manuscript and providing thoughtful and constructive feedback.

However, to be fair I think the authors should state limitation of this study: for example, small sample size, limited medium condition, and unclear molecular mechanisms.

The modified version of the manuscript now mentions in the discussion the limitations pointed out by the reviewer (Line 281).

I am wondering if the authors forgot adding information for the background of model strain MG1655. MG1655 is a K-12 derivative strain that lost F plasmid, thus MG1655 already coevolved with F-type plasmid. It might not grow efficiently without F-type plasmids. How are the plasmids P1-P15 different from the F plasmid? This might be associated with the pattern in Figure 2 top left. I recommend the authors add F plasmid to the Mash distance tree.

The observation we have reported in this paper pertains to differences in fitness costs of plasmids in sensitive and different resistant strains of *E. coli*, which are all derivatives from a single MG1655 background. MG1655 might not grow efficiently without F-type plasmids. Still, if both the resistant and sensitive strains are derived from MG1655, the strain background history may not explain the differences in plasmid-mediated costs in sensitive and resistant backgrounds we observe here. We could add the F plasmid to the mash distance tree, but we are unsure how we can justify that in the paper in a way that will not confuse the readers, especially because we find that the plasmids that do exert costs in sensitive background do not cluster in the Mash distance tree. We have made the tree as suggested by the reviewer (please find it below), but we cannot think of a justifiable way to communicate why we chose to include F plasmid in the tree over all the other plasmids that have not been tested here.

Below I listed other points that need clarification or correction.

Line 91: MG1655 "derivative" strains. "derivative" should be added.

Done (Line 94).

Line 91: I think that strain table containing full stain list rather than table 2 is needed for clarity.

The purpose of Table 2 is to provide a brief overview of the resistance strains and mechanisms we have used for the experiments and we feel including details of sensitive strains will make the table confusing while introducing a lot of unwanted white space. However, we

acknowledge the need to include a list of all the strains used in the paper and we have now included that in the supplementary material (Table S2).

Table 1: family name and allele number of bla genes in subscript should not be italicised.

We thank the reviewer for pointing this out.

Table 1: I think that this table requires more information for each plasmid, especially plasmid group that each plasmid belongs. Please try Mob-typer for MOB group (<https://github.com/phac-nml/mob-suite>) or COPLA for PTU (<https://bmcbioinformatics.biomedcentral.com/articles/10.1186/s12859-021-04299-x>;<https://castillo.dicom.unican.es/copla/>). Otherwise some readers may not show interests to the findings of this study.

We thank the reviewer for informing us about COPLA. We had tried using MOB, but it could not access the local database for plasmids to enable the typing for reasons we do not understand. The web interface for COPLA made plasmid typing much easier. We have determined the PTU for each plasmid and incorporated it in Table 1.

Line 101: Please describe the exact name of delta dapA strain. BW29427 or beta3914? Please update table 2 to include all strains used.

The delta *dapA* strains containing the respective plasmids were in-house constructs that does not bear either of the names the reviewer suggested. We tested the phenotype for the deletion for each strain before we initiated the conjugations. We have modified the methods section to make this more explicit (Line 107).

Line 118: It is not clear why 32DegC was used instead of 35DegC (CLSI guideline) or 37DegC (optimal condition for E.coli).

The decision to use this temperature is purely a practical one. The plasmids being tested here are part of a set of experiments for another project where we are looking at the role of the plasmids in a multispecies environment, where the temperature of 32°C makes it more amenable for the environmental bacterial samples we are using. Thus, the observations reported here were also made at that temperature. Since we do not think the temperature would alter the observations reported here, we decided to report it as is. As a side note—since *E. coli* normally in its life cycle grows at several different temperatures in the environment and in hosts we do not think there is really a “correct” temperature to do this type of experiment (even though 37 °C is the temperature typically used).

Line 119: "g"lucose.

Thank you for pointing this out.

Line 133: Is "bfp" identical to "cfp"? "cfp" is more common, I think.

We do not really understand this comment. Bfp and cfp are two different fluorescent markers (blue and cyan, respectively) that we use to genetically tag the strains for competition experiments.

Line 144: The relative fitness "(Nu)" of. The relative fitness is often expressed by "W" in evolutionary biology. The readers might think why Nu.

The relative fitness is expressed as “v” here and is our preferred metric when relative fitness is determined using initial and final frequencies of competing populations and we have used the same for our other publications as well. We think “W” is apt when initial and final population sizes are available for each of the competing strains, which we are not determining in this case.

Line 171, 179, and Figure 1 legend: this "one-sided t-test" should be "one sample t-test". Since author cannot have a hypothesis for the results (positive effect or negative effect on fitness), two-tailed test is more appropriate. Therefore, "two-tailed one-sample t-test" is more appropriate.

We thank the reviewer for pointing this out. We have made the required corrections.

Line 188: I think that the readers might want a brief introduction of these plasmids regarding the plasmid groups they belong.

The plasmids used here have been studied and described elsewhere and we have included all the relevant information currently available to us in text and table 1.

Table 2: Please list all strains including non-resistant strains used. Otherwise, the method part is difficult to track.

We have included a supplementary table (Table S2) to list all the strains used in the paper.

Line 227/Table 1: It would be nice for readers to see gene content variation among plasmids in Roary-type presence/absence matrix as supplementary figure or main figure.

Roary did not generate a matrix when we fed the sequence data to it. We believe this is because annotations for the sequences submitted to GenBank are generated by the NCBI Prokaryotic Genome Annotation Pipeline, which gave homologous genes across the plasmids different gene names. Since we do not possess the skill set to resolve such issues with Roary, we decided not to use the output generated by Roary or try to resolve it. We do agree with the reviewer that such an addition would indeed be useful, however, we believe the information would not fundamentally alter any of the conclusions in the paper.

Line 228: Please show scale bar for the mash distance tree in figure 4.

We have added a scale bar.

Line 228: Please show Plasmid ID in addition to Accession number in figure 4.

We have added the Plasmid ID.

Line 270: clinically relevant mutation. This must be one of so-called "QRDR mutations" in clinical microbiology. Please state so, and cite one relevant reference.

We have added the information and added a reference to that effect (Line 289).

Line 270: gyrA should be italicized.

Thank you for pointing this out.

Reviewer #2 (Comments for the Author):

Ramith et al. present a specific issue about the interaction between resistance plasmids and bacterial hosts. They demonstrated that *chromosomal resistance mutations could mitigate fitness costs conferred by multi-resistant clinical plasmids. This study is very interesting, and I just have several concerns.*

We thank the reviewer for the positive comments and thoughtful suggestions to improve the manuscript.

*1. What is the prevalence of chromosomal resistance mutations ($\Delta nfsAB$, *gyrA* S83L, and *rpsL* K42N) in clinical isolates, such as *E. coli* and *K. pneumoniae*? Are there disparities in the prevalence of resistance plasmids among clinical isolates with or without chromosomal resistance mutations?*

Among the three resistance mutations tested here, $\Delta nfsAB$ is relevant among UTI patients and is generally not found to be associated with MDR plasmids. *rpsL* mutation is not clinically relevant for *E. coli* and *K. pneumoniae*. *gyrA* S83L is a clinically relevant mutation that is

prevalent among patients treated with ciprofloxacin. Our results do point to the possibility of increased prevalence of resistance plasmids among clinical isolates with these chromosomal mutations, but we are not aware of any study reporting the same. It would indeed be very interesting to find if such an association exists among clinical isolates, but determining this we think is beyond the scope of this study.

2. Clinical isolates harboring resistance plasmids typically undergo numerous generations of co-evolution, whereby the fitness cost associated with these plasmids may be alleviated through the modulation of gene expression. Following the transfer of plasmids into the recipient strain as described in this study, did the authors attempt successive passages of the strains before assessing the fitness cost?

Fitness costs of plasmids can indeed be ameliorated through modulation of gene expression or even through rapid emergence of compensatory mutations on the chromosome or on the plasmid, both of which we wanted to avoid. For this reason, we have avoided multiple serial transfers or multiple rounds of growth in our media before growth is assessed. Both sensitive and resistant hosts were treated the same way and hence any difference in plasmid-mediated fitness costs across replicates is unlikely to be a result of differential evolutionary (or mutational) response in resistant strains.

Reviewer #3 (Comments for the Author):

The authors measure the fitness effects of 15 different MDR plasmids in sensitive E. coli and E. coli which harbour chromosomal mutation providing resistance to various antibiotics to determine if host resistance impacts the costs of plasmid carriage. The authors find that MDR plasmids are more likely to impose significant costs in sensitive hosts relative to resistant hosts. This results in a competitive advantage to resistant strains carrying MDR plasmids relative to sensitive strains harbouring the same plasmids. The authors find that plasmids that did not impose a cost in the sensitive background were genetically closely related, whereas plasmids that imposed costs were genetically diverse. In ciprofloxacin resistance background these same plasmids provided fitness benefits. There was no clear correlation with plasmid size and fitness effects of the plasmids.

We thank the reviewer for carefully reading the manuscript and providing meaningful suggestions to improve the same.

General comments:

Is it not clear that the antibiotic sensitive, NIT-R and constructed ciprofloxacin and streptomycin resistant strains are isogenic, originating from the same parental strain. If they are not isogenic variants other than the resistance mutation may be influencing the fitness in the presence of the plasmids. If not isogenic it is impossible to state if the presence of the chromosomal resistance have anything to do with the observed effects of the plasmids in these backgrounds.

The CIP-R and STR-R strains are constructed from the strain parental to the sensitive and NIT-R strains, so they are not strictly isogenic. However, the only difference between the parental strains of the CIP-R strains and STR-R strains and the sensitive strains is the fluorescent proteins, which are also present in the NIT-R strains (please see the newly added Table S2). We would like to highlight that the growth measurement in each instance is relative to the respective plasmid-free host and hence our conclusion of plasmids having no fitness cost in antibiotic resistance backgrounds will still hold.

Mutations that ameliorate the cost of plasmids can happen very rapidly (see, DOI: 10.1099/mic.0.000862). Can you confirm this did not occur during the conjugation of the clinical plasmids into the E. coli strains? Given that the chromosomal resistant strains already harboured costs, the selective pressure to ameliorate the plasmids may have been even greater.

The only way to confirm this would be through whole genome sequencing. However, since the ameliorations of fitness cost are seen with a wide range of plasmids, we do not think it can be simply explained by mutations occurring during conjugation and hence we do not see the merit in undertaking an investment in that direction. Additionally, the resistance mutations we chose do not have high fitness costs in minimal glucose media (which we have now determined and have been included in the manuscript (Line 198 – 203)). Hence, we believe it is improbable that the difference in plasmid-mediated costs in sensitive and resistant backgrounds is because of mutations that arose post conjugation. Finally, both sensitive and resistant strains go through the same number of growth steps following conjugation to avoid the differential evolutionary responses to plasmid carriage.

Only the streptomycin resistance mutation harbours a cost (~25% reduction in growth rate) among the three chromosomal mutations used in this study. The reviewer raises an important point about compensatory mutations that could explain our observations, but in the absence of significant cost of the resistance mutation for two of the mutations on our media conditions, it is highly improbable that mutations post-conjugation would explain the observations. Notably, plasmid bearing STR-R strains were not found to be fitter than the host (which would be outcome in case). Additionally, we endeavoured to keep the number of generations to a minimum to avoid evolutionary response following plasmid introduction across different biological replicates.

I would have liked to have seen the growth of each of the E. coli backgrounds relative to each other, how much of a fitness effect do the chromosomal resistance mutation impose on the hosts in the absence of the plasmids? This will help to set the results of the effect of the plasmid in context. Do the plasmids have little effect on growth of the resistant strains because they are already very unfit?

Since growth rate determinations for the four host strains were conducted on different days, a post-hoc comparison of the growth rates may not be accurate and that is why we did not include that in the initial submission. Following the suggestion by the reviewer, we estimated the growth rates of the four host strains along with wildtype MG1655 in a separate experiment and the corresponding values have been included as a supplement figure in the manuscript (Fig S1) and briefly summarised in the manuscript (line 202). As noted before, and mentioned in the discussion section of the manuscript, only the STR-R mutant is relatively unfit. However, since all the comparisons are with the plasmid-free host and not the parent strain, any plasmid-mediated effect on fitness should still be apparent.

What is the correlation in fitness effects of the plasmids across the different backgrounds? Are the same plasmids more likely to impose fitness benefits across all resistant backgrounds?

As reported in Fig. 2, none of the plasmids showed any costs in any of the resistant backgrounds and some even had positive fitness effects only in CIP-R background. Thus, we are not able to predict if any of the plasmids are more likely to impose any cost or provide any benefit across all resistant backgrounds. Further research will throw light into the mechanistic aspects of interactions between resistance mutations and plasmid carriage that could explain the observations reported here. To make it easier to view this aspect of the data, we have made a separate figure and included it as a supplementary figure (Fig S2).

Could other properties of the MDR plasmids correlate with fitness (e.g. number of resistance genes, of the presence of specific resistances?)

The central observation of the paper is the amelioration of fitness costs in resistant backgrounds under nutrient-limited conditions and not specific mechanistic aspects of plasmid costs, which has been dealt with very ably by other competent research groups over the past few years. The genetic and molecular aspects of plasmid costs in nutrient-limited conditions under different resistant backgrounds would indeed be a very interesting topic for further research but we think this is beyond the scope of our current manuscript.

Specific comments:

Line 48, plasmids can or may carry other genes

We have modified the sentence accordingly.

Line 190-192, are the strains containing chromosomal mutations providing resistance to NIT, CIP and STR isogenic with the drug sensitive MG1655?

CIP-R and STR-R strains are isogenic to MG1655 but for the resistance mutation, while NIT-R and sensitive strains used here express fluorescent proteins (YFP and BFP). Plasmid mediated fitness costs are determined by comparing the growth rates of plasmid bearing strain with its plasmid-free hosts and we do not draw any conclusion from the comparison of the growth rates across resistance backgrounds, hence our observation of differential costs of plasmid carriage among sensitive and resistance strains still holds. The competition among sensitive and NIT-R strains is conducted reciprocally between fluorescently labelled isogenic strains to overcome the differential costs, if any, of the fluorescent proteins themselves.

Line 214-215, The selection criteria does not seem to correlate with the plasmids selected: states that they selected the plasmids with the highest exponential growth rates in the NIT-R strain (P5, P8, P9, P14 and P15) but P5 P8 and P9 in figure 2 do not have a significant increase in growth rate relative to plasmid free. Why select these over say P1, P2 or P4?

Since none of the plasmids has a significant effect on the fitness of NIT-R strains, we selected the plasmids with the highest observed average relative growth rate for this experiment, though they are not statistically different from 1. We needed some criteria to select a subset of plasmids for this experiment and this is the one we chose.

Line 218, P8, if it's not significant then it did not have a fitness advantage.

We have modified the sentence (Line 225).

Line 221-222, do you mean relative to P9, P14 and P15 and not out of all the 15 plasmids? They still imposed a significant cost in the sensitive strain.

Yes, we meant among the strains selected for this experiment and this has been made more explicit (Line 240).

Figure 2, why isn't data available for P15 in STR-R strain?

We were unable isolate transconjugants for that plasmid in STR-R background. This has been added to the methods section now (Line 120).

Figure 4, put the plasmids ids used in the other figures on the nodes of the tree to make it easier to compare between data sets. The three needs a scale.

We thank the reviewer for these suggestions. We have made the required additions.

Re: Spectrum04206-23R1 (Antibiotic resistance begets more resistance: Chromosomal resistance mutations mitigate fitness costs conferred by multiresistant clinical plasmids)

Dear Dr. Ramith R Nair:

I am pleased to inform you that your manuscript has been accepted for publication. Your revised manuscript has been reviewed by two reviewers. They recommended acceptance of your work.

Your manuscript has been accepted, and I am forwarding it to the ASM production staff for publication. Your paper will first be checked to make sure all elements meet the technical requirements. ASM staff will contact you if anything needs to be revised before copyediting and production can begin. Otherwise, you will be notified when your proofs are ready to be viewed.

Sincerely,
Sen Pei
Editor
Microbiology Spectrum

Reviewer #1 (Comments for the Author):

The manuscript has been improved

Reviewer #3 (Comments for the Author):

The responses and changes to the manuscript have addressed all of my original comments, I have no further concerns regarding the manuscript.